# Deep Instruction Tuning for Segment Anything Model

## Xiaorui Huang
Key Laboratory of
Multimedia Trusted
Perception and Efficient
Computing, Ministry of
Education of China,
Xiamen University
Xiamen, Fujian, China
huangxiaorui@stu.xmu.edu.cn

## Gen Luo
Key Laboratory of
Multimedia Trusted
Perception and Efficient
Computing, Ministry of
Education of China,
Xiamen University
Xiamen, Fujian, China
luogen@stu.xmu.edu.cn

## Chaoyang Zhu
The Department of
Computer Science and
Engineering, The Hong
Kong University of Science
and Technology
HongKong, China
sean.zhuh@gmail.com

## Bo Tong
Key Laboratory of
Multimedia Trusted
Perception and Efficient
Computing, Ministry of
Education of China,
Xiamen University
Xiamen, Fujian, China
tongbo@stu.xmu.edu.cn

## Yiyi Zhou[†]
Key Laboratory of
Multimedia Trusted
Perception and Efficient
Computing, Ministry of
Education of China,
Xiamen University
Xiamen, Fujian, China
zhouyiyi@xmu.edu.cn

## Xiaoshuai Sun
Key Laboratory of
Multimedia Trusted
Perception and Efficient
Computing, Ministry of
Education of China,
Xiamen University
Xiamen, Fujian, China
xssun@xmu.edu.cn

## Rongrong Ji
Key Laboratory of
Multimedia Trusted
Perception and Efficient
Computing, Ministry of
Education of China,
Xiamen University
Xiamen, Fujian, China
rrji@xmu.edu.cn

## Abstract

Recently, *Segment Anything Model* (SAM) has become a research hotspot in the fields of multimedia and computer vision, which exhibits powerful yet versatile capabilities on various (un) conditional image segmentation tasks. Although SAM can support different types of segmentation prompts, we note that, compared to point- and box-guided segmentations, it performs much worse on text-instructed tasks, *e.g.*, *referring image segmentation* (RIS). In this paper, we argue that deep text instruction tuning is key to mitigate such shortcoming caused by the shallow fusion scheme in its default light-weight mask decoder. To address this issue, we propose two simple yet effective *deep instruction tuning* (DIT) methods for SAM, one is end-to-end and the other is layer-wise. With minimal modifications, DITs can directly transform the image encoder of SAM as a stand-alone vision-language learner in contrast to building another deep fusion branch, maximizing the benefit of its superior segmentation capability. Extensive experiments on three highly competitive benchmark datasets of RIS show that a simple end-to-end DIT can improve SAM by a large margin, while the layer-wise DIT can further boost the performance to state-of-the-art with much less data and training expenditures. Our code is released at: https://github.com/wysnzzzz/DIT.

[†]Corresponding Author.

## CCS Concepts

• **Computing methodologies** → **Image segmentation**; **Scene understanding**.

## Keywords

Referring Image Segmentation, Deep Instruction Tuning, Segment Anything Model

**ACM Reference Format:**
Xiaorui Huang, Gen Luo, Chaoyang Zhu, Bo Tong, Yiyi Zhou, Xiaoshuai Sun, and Rongrong Ji. 2024. Deep Instruction Tuning for Segment Anything Model. In *Proceedings of the 32nd ACM International Conference on Multimedia (MM '24), October 28-November 1, 2024, Melbourne, VIC, Australia*. ACM, New York, NY, USA, 10 pages. https://doi.org/10.1145/3664647.3680571

## 1 Introduction

Motivated by the great success of *large language models* (LLMs) [59, 65], Kirillov *et al.* [28] recently propose a *Segment Anything* project to build a foundation model for image segmentation, termed *Segment Anything Model* (SAM). SAM demonstrates an impressive yet versatile segmentation capacity by pre-training on over one billion masks. Moreover, it can support interactive segmentation conditioned on various input prompts [28], such as *points*, *boxes* or *texts*. Its outstanding ability also arouses great interest from the communities of multimedia [4, 5, 13, 85] and computer vision [6, 8, 71], and has been recently transferred to various segmentation tasks, such as *medical image segmentation* [15, 61, 76], *urban image segmentation task* [88], and *marine animal segmentation* [87], or to provide instance masks for image editing tasks [13, 77].

However, the text-awareness of SAM is much inferior than following the prompts of points or boxes [28]. For instance, on *true-color segmentation* task [75], SAM can achieve state-of-the-art performance of 85.2% *mIoU* with box guidance. In stark contrast, SAM

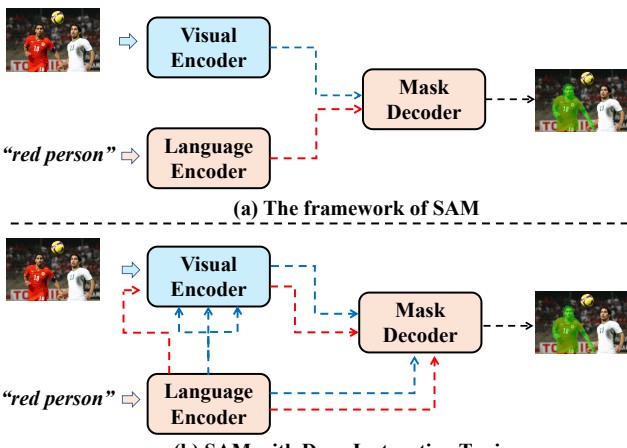

**(a) The framework of SAM**

**(b) SAM with Deep Instruction Tunings**

**Figure 1: Illustration of the default fusion of SAM (a) and our *deep instruction tuning* (DIT) methods (b), *i.e.*, the end-to-end (red) and layer-wise (blue) ones. Compared with the default shallow fusion, DITs regard the visual encoder as an deep multi-modal learner for cross-modal fusion and visual feature learning, thereby achieving the full interactions between text and image for text-guided SAM.**

can only achieve 58.0% *mIoU* when fine-tuned on the RefCOCO benchmark [86] of the popular text instructed task, *i.e.*, *referring image segmentation* (RIS) [86] as shown in Fig. 2. To explain, text instructions naturally exhibit linguistic ambiguities, while location prompts (points and boxes) are precise. Meanwhile, recent Transformer-based vision models [11, 50] have well incorporated with this type of spatial information, *e.g.*, *position embedding*, and it is natural for SAM to well master these location prompts. In addition, the collection of instance-wise text descriptions is more laborious, and even the public *Segment Anything* project also does not release the text-conditioned annotations [28].

Even worse, the shallow fusion scheme in SAM further aggravates such weakness and degrades its multi-modal ability. Concretely, for either point, box or text, input prompt is first encoded by the prompt encoder of SAM, then it is directly fed into the light-weight encoder for mask prediction. In SAM, prompts can only interact to a limited extent with visual tokens via two cross-attention layers in its mask decoder [28]. Although such mechanism is well suited for point and box prompts which are relatively easy to follow, it is insufficient for SAM to infer the intentions behind texts step-by-step. Oftentimes, SAM needs to resolve linguistic ambiguity first through the attributes and relationships of the referent, then the object can be segmented using its powerful and class-agnostic segmentation ability. To this end, we argue that

*"Deep text instruction tuning is essential for SAM."*

To validate this argument, we propose two *deep instruction tuning* (DIT) methods for SAM. Different from existing RIS models [7, 19, 24, 33] that build deep fusion branches upon image encoders, we focus on improving text instruction following ability of SAM without modifying its structure. This property can help DITs make full use of its superior segmentation capability while reducing the training cost to a large extent. In particular, we directly extend

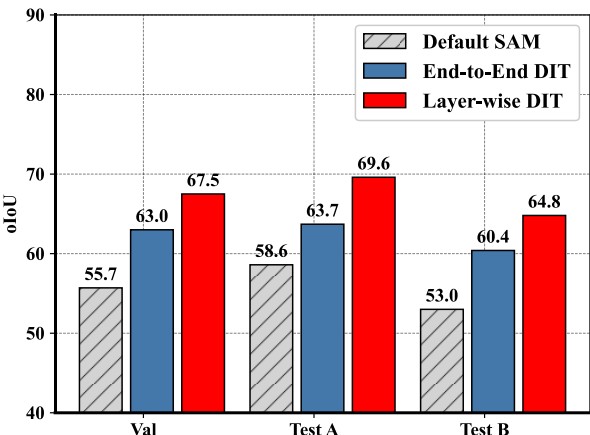

**Figure 2: The comparison among the default fusion scheme of SAM (with ViT-B) and our deep instruction tuning methods on RefCOCO [86], *i.e.*, the end-to-end and layer-wise ones, respectively[1]. In SAM, the text prompts only interactive with the visual tokens in the shallow mask decoder, which is insufficient for cross-modal fusion and text-guided segmentation. Our simple deep tuning methods can improve the performance by a large margin without significantly changing the architecture of SAM.**

SAM to a deep vision-language model. Enabled by the flexibility of Transformer-based encoder [11], we convert text instruction words to input tokens and concatenate them with the visual ones for deep multi-modal fusion and feature learning, which forms the first DIT scheme in this paper, we termed *end-to-end DIT* (E-DIT), as depicted in Fig. 3. This simple method improves SAM by a large margin, *e.g.*, +11.8% on RefCOCO *val*, as shown in Fig. 2. To further enhance instruction learning, we also propose a layer-wise instruction tuning method (L-DIT) that independently projects and inserts the text words into each layer of SAM. This layer-wise method has a similar principle with *soft prompt tuning* [20]. The difference is that the inserted tokens are the projected text embeddings that vary across examples, instead of task-specific and parameterized prompt tokens that are static. Besides, DITs aim to project the text words onto the well-learned visual semantic space of SAM, while prompt tuning is often to help the model better adapt the downstream task without full tuning [20].

With the proposed DIT methods, we apply SAM to the popular text-guided *Referring Image Segmentation* (RIS) [44] task, and term the new models as *DIT-SAMs*. Extensive experiments are conducted on three highly competitive RIS benchmarks, namely RefCOCO [86], RefCOCO+ [86] and RefCOCOg [56, 58]. The experimental results not only show the great advantages of *DIT-SAMs* over the default SAM, *e.g.*, up to 11.6% improvement on RefCOCO *testA*, but also confirm the competitiveness of DIT-SAM as a stand alone RIS model. Overall, we provide a fast and feasible solution to extend SAM's modal-modal ability.

To summarize, our contributions are three-fold:

- We reveal the key drawbacks of SAM in terms of text-guided segmentation and argue that deep text instruction tuning is key to success.

---

[1]The text-prompt version of SAM is not released, so we follow its setting to initialize the text encoder and tune on RefCOCO. ViT is frozen here.

- We propose two simple yet effective *deep instruction tuning* (DIT) methods for SAM, which are end-to-end and layer-wise, respectively. These DIT methods pave a quick and feasible way for the multi-modal extension of SAM.
- The proposed DIT-SAMs not only greatly outperforms the default SAM on RIS benchmarks, but also yield strong competitiveness against existing SOTA methods of RIS.

## 2 Related Work

### 2.1 Image Segmentation

Image segmentation [1, 17, 51, 62] is a fundamental task in multimedia and computer vision, which requires pixel-level understanding of a given image. Recent years have witnessed its fast development supported by a bunch of segmentation-related tasks, such as *semantic segmentation* [1, 2, 17, 51, 78] which categorizes individual pixels into a predetermined set of classes, *instance segmentation* which primarily concerns the recognition and delineation of distinct object instances [17, 64, 68], and *panoptic segmentation* which merges semantic and instance segmentation tasks by assigning class labels and instance identification [27, 36]. In addition, a set of advanced segmentation networks [17, 55, 64, 68, 94] are also proposed to promote the development of image segmentation. For instance, He *et al.* [17] propose Mask-RCNN for object instance segmentation. SOLO [67] converts instance segmentation into a single-shot classification-solvable problem. IFR [55] is a implicit feature refinement module for high-quality instance segmentation, and TAR [94] presents the first attempt of weakly-supervised text instance segmentation by text recognition and segmentation. Recently, the *Segment Anything* project [28] proposes a powerful and general model for image segmentation, termed *segment anything model* (SAM). This project includes the creation of an unprecedentedly large segmentation dataset, so that SAM is trained with about 1 billion image-mask pairs. Via large-scale pre-training, SAM exhibits robust and versatile performance across various segmentation challenges which can also handle different types of prompts for conditional segmentation, such as *point*, *box* and *text*. More recently, SAM has been widely introduced to a set of segmentation-related tasks, such as *low-level structural segmentation* [3], *surgical scene segmentation* [60], and *medical image segmentation* [15, 61, 76]. However, these applications mainly make use of the interactive capability of SAM on point or box prompts, while the text-instructed application is barely explored. Moreover, SAM also works much worse with texts than the other types of prompts due to the complexity of text information and the insufficient fusion of its default shallow decoder. In this paper, we propose deep instruction tuning for the multi-modal enhancement of SAM.

### 2.2 Referring Image Segmentation

*Referring image segmentation* (RIS) [86] is one of the most popular text-conditioned segmentation task [21, 47, 53, 54, 63, 92], which grounds the referent with binary masks according to the given natural language expression. Early research in RIS [43, 86] typically adopts a two-stage pipeline and frames RIS as a region-text matching problem. One-stage RIS models [35, 42, 57] recently garner growing attention from both academia and industry. In these one-stage models, text features are often embedded into the vision

network and directly output the mask of the referent. Inspired by the success of Transformer, various Transformer-based RIS models [18, 39, 45, 72, 79, 92] are recently proposed to improve the multi-modal reasoning ability for RIS. VLT [10] and SeqTR [92] use a encoder-decoder framework to fuse visual and language inputs into the decoder. ReSTR [24] employs two separate Transformer encoders to handle two different modalities. More recently, attempts have also been made to explore a wider range of text conditional segmentation by combining large language models (LLMs) and training with large-scale image-text pairs [30, 49, 81, 93]. For instance, UniLSeg [49], LISA [30] and SEEM [93] focus on text-instructed open-world segmentation, which typically require massive multi-modal training data and LLMs like LLaMA-7B [65]. Compared with these advancements, we mainly focus on addressing the text awareness of SAM and only conduct quick validations on RIS benchmarks. Among these progresses, the most relevant works to our DITs are LAVT [83] and ETRIS [34], which also only use two modality encoders for deep multi-modal fusion. However, our DITs greatly differ in both principle and designs. In particular, LAVT and ETRIS aim to achieve multi-modal interactions via cross-attention-based modules between two Transformers. In contrast, the principle of DITs is to project the text words onto the semantic space of SAM's image encoder, thus achieving feature fusion and learning in one forward network. This property also makes DIT's designs more intuitve and simpler, which also exhibits obvious merits in both training costs and performance.

### 2.3 Prompt Tuning

With the ever increasing parameter size of pre-trained models [9, 11, 65], the direct fine-tune on downstream tasks becomes prohibitively expensive. In this case, *prompt tuning* [14, 16, 48, 70] is proposed to use hand-craft or learnable prompt tokens to mitigate distribution shifts between pre-training and downstream tasks. The significant achievements of prompt tuning in natural language processing (NLP) have also inspired its application to computer vision and vision-language research [46, 52, 74, 90, 91]. CoOp [91] remains CLIP parameters unchanged and uses learnable vectors as soft prompts for the input text, thereby maximizing CLIP's zero-shot retrieval capabilities. To address the overfitting issue on base classes, CoCoOp [90] proposes instance adaptation by incorporating the visual features of each image into the parameterized prompts. RPO [31] enhances the robustness of VL models by masked attention to prevent the added learnable embeddings from altering the original internal representations of the model. More recently, Wu *et al.* [74] propose a method of *approximated prompt tuning* to reduce the additional cost of the inserted prompt tokens. Regarding soft prompt tuning [32, 52, 73], whether applied deeply or shallowly, it employs learnable tokens to assist the model in adapting to downstream data distributions without the necessity for complete tuning. Although the proposed DIT methods and prompt tuning are quite similar in process, but they still hold different principles and designs. To explain, prompt tuning often adopts learnable or hand-craft tokens to brdige the gap between pre-training and downstream tasks, which are often static to tasks [20]. In contrast, DITs aim to project the text tokens onto the semantic space of SAM, which are dynamic for different examples. Conclusively, prompt tuning primarily facilitates

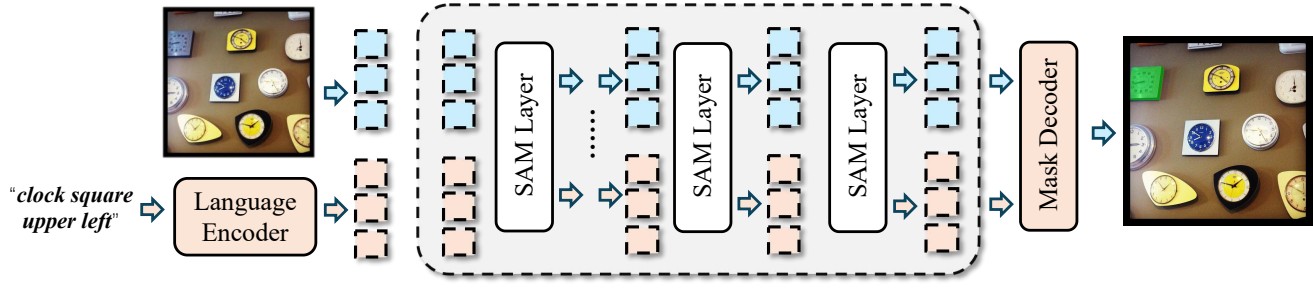

**(a) End-to-end Deep Instruction Tuning for SAM**

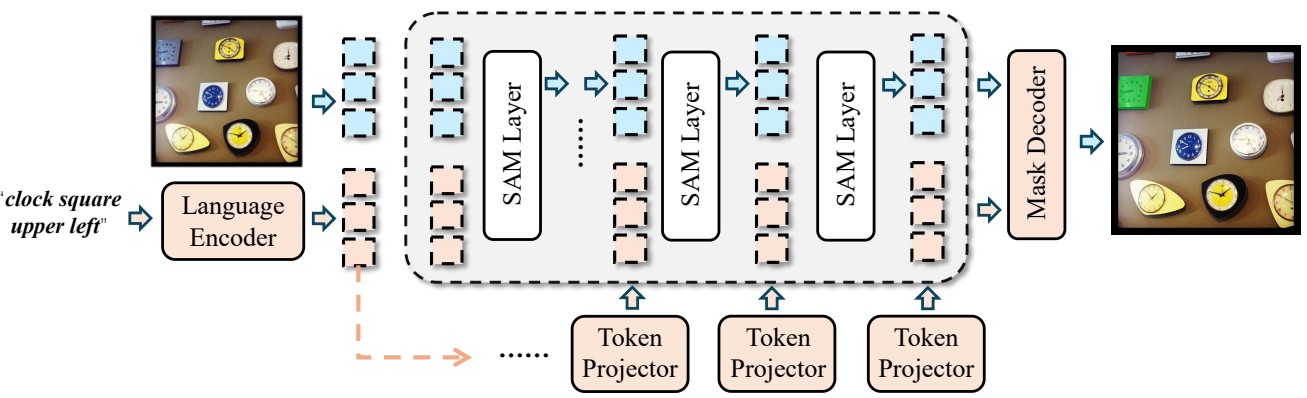

**(b) Layer-wise Deep Instruction Tuning for SAM**

**Figure 3: Overall pipeline of Deep Instruction Tuning (DIT). We propose (a) End-to-end DIT (E-DIT) that appends language instructions before visual features to achieve early-fusion for enhancing multi-modal reasoning capability; (b) Layer-wise DIT (L-DIT) that layer-wisely projects to text tokens interact with the existing sequences in the well learned semantic space.**

rapid task adaptation, whereas our DITs offer a novel approach to VL reasoning for SAM.

## 3 Method

### 3.1 Problem Definition

*Referring image segmentation* (RIS) [86] aims to segment the referred object in an image $I$ conditioned on the paired natural language expression $T$, its objective is formulated as

$$\arg \min_{\theta} \mathcal{L}(\mathcal{F}(I, T; \theta), M), \tag{1}$$

where $\mathcal{F}$ denotes the model with its parameters $\theta$, $M \in \mathbb{R}^{H \times W}$ is the ground-truth mask, and $\mathcal{L}$ is the canonical *cross entropy* loss.

To achieve this task, SAM [28] first extracts visual and text features through its vision and language encoders, based on which the light-weight decoder takes as input to predict the binary masks. Thus, the model $\mathcal{F}$ instantiated by SAM is

$$\mathcal{F}(I, T; \theta) = \mathcal{F}_d\big(\mathcal{F}_v(I), \mathcal{F}_t(T)\big), \tag{2}$$

where $\mathcal{F}_v$, $\mathcal{F}_t$ and $\mathcal{F}_d$ refer to visual backbone, language encoder and mask decoder, respectively.

The mask decoder is very lightweight and only contains two cross-attention layers, which are in charge of both mask decoding and multi-modal interactions. In this case, SAM can be regarded as one of the late-fusion models which proved to be less effective than

the early-fusion ones [44, 53, 54, 92] in addressing the complicated multi-modal reasoning tasks. Meanwhile, due to the lack of training on massive cross-modal data, text-guided SAM performs worse than the point- or box-guided SAM, leading to much inferior cross-modal abilities than existing SOTA methods in RIS [44, 92].

We introduce *Deep Instruction Tuning* (DIT) for SAM to better follow the intentions reflected in the natural language expression $T$. In particular, DIT keeps the whole SAM architecture unchanged and directly uses the image encoder as a strong multi-modal fusion network. The text instruction words can continually interact with the visual contents through the whole feature learning process, helping SAM progressively locate the referent. Specifically, SAM with DIT can be then formulated as

$$\mathcal{F}(T, I; \theta) = \mathcal{F}_d\big(\mathcal{F}_v(I, \phi(\mathcal{F}_t(T))), \mathcal{F}_t(T)\big). \tag{3}$$

Here, $\phi(\cdot)$ is a mapping function to project the text features onto the default visual space of SAM.

### 3.2 End-to-end Deep Instruction Tuning

One straightforward solution for SAM is to directly feed the instruction words to its ViT-based image encoder. Due to the flexibility of Transformer architecture, the text features can be also regarded as additional input tokens for feature transformations. This attempt is also validated in previous VL works like ViLT [25], and has a similar process with the popular prompt tuning [20].

**Table 1: Comparison of different text-instructed schemes on RefCOCO. *Default* refers to the default lightweight fusion scheme of SAM. *Add dec layers* is to increasing the depth of the mask decoder. * denotes that the image backbone is frozen, while the text and mask decoders are tuned. The best and second best performances are in bold and underlined.**

| Method | val | | | | | testA | | | | | testB | | | | |
|---|---|---|---|---|---|---|---|---|---|---|---|---|---|---|---|
| | P@0.5 | P@0.7 | P@0.9 | oIoU | mIoU | P@0.5 | P@0.7 | P@0.9 | oIoU | mIoU | P@0.5 | P@0.7 | P@0.9 | oIoU | mIoU |
| Default | 72.20 | 59.56 | 18.78 | 60.74 | 63.41 | 75.28 | 63.76 | 18.76 | 62.95 | 65.65 | 66.95 | 52.63 | 19.82 | 56.69 | 59.95 |
| Default* | 64.77 | 51.92 | 16.82 | 55.67 | 57.98 | 68.29 | 56.47 | 16.86 | 58.56 | 60.85 | 61.32 | 46.66 | 17.35 | 52.98 | 55.74 |
| Add 2 dec layers | 80.80 | 71.65 | 27.03 | 67.17 | 70.51 | 83.52 | 75.27 | 25.48 | 69.63 | 72.48 | 76.57 | 65.19 | 29.24 | 64.04 | 67.61 |
| Add 2 dec layers* | 71.44 | 59.03 | 18.61 | 59.88 | 62.88 | 75.34 | 63.06 | 18.29 | 62.77 | 65.55 | 66.48 | 52.41 | 19.78 | 56.79 | 59.80 |
| Add 4 dec layers | 79.58 | 70.21 | 26.45 | 65.34 | 69.51 | 82.27 | 73.68 | 25.39 | 67.82 | 71.44 | 74.23 | 63.78 | 28.28 | 62.28 | 66.02 |
| Add 4 dec layers* | 71.37 | 58.79 | 18.52 | 59.46 | 62.54 | 74.26 | 63.07 | 19.12 | 62.05 | 65.02 | 65.48 | 51.43 | 19.60 | 55.62 | 58.86 |
| End-to-end DIT | 81.78 | 73.07 | 28.27 | 68.07 | 71.46 | 85.28 | 77.17 | **27.56** | 70.81 | 73.90 | 78.02 | 67.60 | 30.20 | 65.58 | 69.36 |
| End-to-end DIT* | 76.51 | 66.00 | 23.19 | 62.96 | 66.87 | 78.89 | 68.41 | 21.31 | 63.74 | 67.83 | 72.90 | 60.01 | 25.39 | 60.43 | 64.63 |
| Layer-wise DIT | **85.68** | **76.61** | **29.89** | **71.98** | **74.73** | **88.06** | **79.17** | 25.94 | **74.51** | **75.62** | **81.28** | **69.11** | **30.95** | **68.77** | **71.21** |
| Layer-wise DIT* | 81.22 | 70.21 | 24.31 | 67.50 | 69.97 | 83.89 | 74.66 | 24.36 | 69.55 | 72.44 | 77.26 | 65.44 | 28.26 | 64.77 | 68.12 |

Here, we term this DIT solution as the end-to-end tuning, abbreviated as E-DIT. Concretely, given a text instruction $T$, we first use the language encoder, *i.e.*, BERT [9], to extract its representations, denoted as $F_t \in \mathbb{R}^{l \times d}$, where $l$ denotes the length of instruction. The visual features after the patch embedding layer are denoted as $F_v^0 \in \mathbb{R}^{m \times d}$. Here, $m = H \times W/P^2$ is the number of patches, where $H \times W$ is the image resolution and $P^2$ is the patch size. Afterwards, we use a linear layer to project $F_t$ onto the visual semantic space of SAM, and obtain $F_t^0$ via

$$F_t^0 = F_t W_0 + b_0, \tag{4}$$

where $W_0 \in \mathbb{R}^{d \times d}$ and $b_0 \in \mathbb{R}^d$ denote the projection weight and the bias term, respectively.

These text tokens and image patches are then processed by the following Transformer layers of the encoder for both image feature learning and cross-modal fusion. The output image features $F_v^n$ alone with the text features $F_t$ are fed to the lightweight mask decoder. In this case, E-DIT is formulated by

$$F_m^0 = [F_t^0; F_v^0], \tag{5}$$

$$F_m^{i+1} = L_i(F_m^i), \tag{6}$$

$$M' = \mathcal{F}_d(F_v^n, F_t W_o). \tag{7}$$

Here, $L_i$ denotes the $i$-th Transformer layer of visual backbone, and $W_o \in \mathbb{R}^{d \times d}$ is the projection weight. $[\cdot]$ denotes concatenation. $M'$ is the predicted binary mask.

Considering the case of freezing SAM's backbone, Eq. 4 only adds a few parameters for semantic projection. When the hidden Transformer layers are frozen, the inserted text tokens do not fit well into visual space. For the output of each layer, we also add a linear projection to facilitate text adaption.

### 3.3 Layer-wise Deep Instruction Tuning

According to Wu *et al.* [74], it is often difficult for additionally inserted tokens to fully interact with the existing feature sequences

in the well learned semantic space, especially under the end-to-end manner. In this case, we also propose an effective layer-wise instruction tuning method for SAM, demoted as L-DIT.

Concretely, we project the text tokens onto the sub-semantic space of each Transformer layer of SAM' encoder, thereby mitigating the modality gap for better tuning. Given the text features $F_t$, we linearly insert them into each layer of the image encoder, *i.e.*, $F_t^i$, and the input sequence is defined by

$$F_m^i = [F_t^i; F_v^i], \tag{8}$$

where $i$ denotes the $i$-th layer.

Similar to Eq. 5-7, the final output visual tokens $F_v^k$ as well as the projected text features $F_t'$ are fed to the lightweight decoder for mask prediction. Compared with E-DIT, this layer-wise scheme can better project text tokens onto the subspaces of SAM's encoder, thereby improving the efficiency of text instruction tuning.

## 4 Experiments

To validate the proposed DITs, we conduct extensive experiments on three highly competitive benchmarks of RIS, *i.e.*, RefCOCO [86], RefCOCO+ [86] and RefCOCOg [56], and also compare L-DIT with a bunch of state-of-the-art (SOTA) methods [40, 44, 83, 92] in RIS.

### 4.1 Datasets and Metrics

**RefCOCO** [86] dataset is collected in an interactive two-player game [23]. There are 142,210 referring expressions and 50,000 segmentation masks in 19,994 images collected from COCO [38]. It is divided into *train*, *val*, *testA*, and *testB* with 120,624, 10,834, 5,657, and 5,059 samples, respectively.

**RefCOCO+** [86] contains 141,564 natural language expressions and 49,856 masks in 19,992 images. Compared to RefCOCO, expressions in RefCOCO+ describe more about attributes of the referent, *e.g.*, color, shape, digits, and avoid using words of absolute spatial location such as *left* and *right*.

**RefCOCOg** [56, 58] has 104,560 expressions for 54,822 objects in 26,711 images. The expressions of RefCOCOg contain 8.4 words on

**Table 2: The combination with conventional prompt tuning. Here, *dynamic* refers to the projected text words changed according to different examples, *i.e.*, the setting of DITs, and *static* denotes the learnbale tokens.**

| Setting | Prompt | val | | | | testA | | | | testB | | | |
|---|---|---|---|---|---|---|---|---|---|---|---|---|---|
| | | P@0.5 | P@0.7 | oIoU | mIoU | P@0.5 | P@0.7 | oIoU | mIoU | P@0.5 | P@0.7 | oIoU | mIoU |
| End-to-end | dynamic | 81.78 | 73.07 | 68.07 | 71.46 | 85.28 | 77.17 | 70.81 | 73.90 | 78.02 | 67.60 | 65.58 | 69.36 |
| End-to-end | dynamic+static | 82.60 | 73.89 | 68.53 | 72.11 | 85.76 | 77.76 | 71.21 | 74.12 | 78.41 | 67.39 | 65.64 | 69.38 |
| Layer-wise | dynamic | **85.68** | **76.61** | **71.98** | **74.73** | **88.06** | **79.17** | **74.51** | **75.62** | **81.28** | **69.11** | **68.77** | **71.21** |
| Layer-wise | dynamic+static | 81.09 | 71.46 | 67.31 | 70.84 | 83.83 | 75.30 | 69.54 | 72.78 | 76.84 | 65.70 | 63.97 | 67.98 |

**Table 3: The impact of different text word injections for L-DIT. *CrossAtt* and *FFN* refer to mutli-head cross attention and feed-forward network [66], respectively. Linear projection is the setting of L-DIT.**

| Setting | val | | | | | testA | | | | | testB | | | | |
|---|---|---|---|---|---|---|---|---|---|---|---|---|---|---|---|
| | P@0.5 | P@0.7 | P@0.9 | oIoU | mIoU | P@0.5 | P@0.7 | P@0.9 | oIoU | mIoU | P@0.5 | P@0.7 | P@0.9 | oIoU | mIoU |
| Linear project | **85.68** | **76.61** | **29.89** | **71.98** | **74.73** | **88.06** | **79.17** | 25.94 | **74.51** | **75.62** | **81.28** | **69.11** | 30.95 | **68.77** | **71.21** |
| CrossAtt | 80.49 | 70.70 | 25.63 | 67.06 | 70.33 | 83.30 | 73.45 | 24.43 | 69.67 | 72.01 | 76.98 | 65.33 | 28.08 | 64.85 | 68.13 |
| CrossAtt + FFN | 82.07 | 72.22 | 27.09 | 68.26 | 71.42 | 84.96 | 76.29 | **26.25** | 71.03 | 73.52 | 78.34 | 66.29 | 29.43 | 65.68 | 69.00 |

**Table 4: The impact of tuning on SAM. Here, ✗ means not tuning, and *Mix* denotes the joint training with point, box and text[2]. The metric used is *oIoU*.**

| Setting | Train | Prompt | val | testA | testB | Inference |
|---|---|---|---|---|---|---|
| Default SAM | ✗ | Point | 75.84 | 78.11 | 75.38 | 119ms |
| Default SAM | ✗ | Box | 75.32 | 75.38 | 76.60 | 120ms |
| Default SAM | Text | Text | 60.74 | 62.95 | 56.69 | 123ms |
| Layer-wise DIT | Mix | Point | 85.16 | 85.59 | 84.51 | 123ms |
| Layer-wise DIT | Mix | Box | 87.47 | 87.43 | 87.74 | 123ms |
| Layer-wise DIT | Mix | Text | 69.35 | 71.04 | 66.76 | 123ms |
| Layer-wise DIT | Text | Text | 71.98 | 74.51 | 68.77 | 123ms |

average while that of RefCOCO and RefCOCO+ is only 3.6 and 3.5 words. It includes object appearance and location.

**Metrics.** We use the *overall intersection-over-union* (oIoU), the *mean intersection-over-union* (mIoU), and *precision* at threshold of 0.5, 0.7, and 0.9 as evaluation metrics. The oIoU is measured by the ratio of the total area of intersection to the total area of union across all test samples. The metric favors large objects over small ones. The mIoU is calculated as the average of the IoU values between predictions and the ground truth for all test samples. This measure is applied uniformly to both large and small objects.

## 4.2 Experimental Setups

Since the text-prompt version of SAM is not publicly available, we use the default image encoder and segmentation decoder of SAM, and randomly initialize the text encoder in this paper. For visual backbone, we use ViT-B/16 initialized from pre-trained weights of SAM. BERT with 12 layers and a hidden dimension of 768 is used as language encoder. We keep the aspect ratio and resize the image to $512 \times 512$. Following previous works [53, 92], we apply *color augmentation*, *Gaussian blur* and *random horizontal flipping*

to augment the data. To binarize the prediction of RIS, we set a threshold of 0.35. The optimizer is *Adam* [26] with a learning rate of 1e-4, which is multiplied by a decay factor of 0.2 at the 30-*th* and the 35-*th*, and the batch size is set to 64. The warm-up epoch is configured to 3. The training process spans 40 epochs. More details can refer to our anonymously released project.

## 4.3 Quantitative Analysis

*4.3.1 Comparison with the default scheme.* In Tab. 1, we first compare our *deep instruction tuning* (DIT) methods with the default fusion scheme of SAM and its alternatives, *i.e.*, adding more layers to the mask decoder. We can see that the default fusion scheme performs much worse than DITs on RefCOCO under the setting of either fully tuning or freezing backbone, suggesting that the default lightweight decoder is insufficient in text following. With more decoding layers, this issue can be alleviated to some extent, but the performance still lags behind DITs especially when the image encoder is frozen. To explain, the additional fusion layers can enhance the cross-modal interactions, but also undermines the powerful segmentation capability of SAM pre-trained on massive data. In contrast, without modifying the main structure of SAM, our DIT methods can improve the performance greatly. Another interesting finding from Tab. 1 is that the advantage of layer-wise DIT is more obvious under the setting of freezing backbone. This result confirms our assumption that layer-wise DIT can better help to project text tokens onto the semantic space of SAM. Overall, Tab. 1 well validates the effectiveness of DITs for text-instructed SAM.

*4.3.2 Combination with prompt tuning.* Tab. 2 present the results of combing DITs with conventional the prompt tuning [74, 90, 91], *i.e.*, adding learnable tokens to the text prompts. Under end-to-end tuning, the addition of learnable tokens can improve performance to a certain extent, mainly due to the increase of the parameter capacity of DIT for better text tuning. However, these prompt tokens are

---

[2]Boxes and points are only processed in the default decoder.

**Table 5: Comparison with the state-of-the-art methods on three RIS datasets. Here, *Scratch* refers to only using the RIS examples of these datasets for training, while *Pre-trained* denotes the use of a large number of visual grounding examples [29] for pre-training. The metric used is *oIoU*.**

| Method | Visual Backbone | Textual Encoder | Pretraining Data | RefCOCO | | | RefCOCO+ | | | RefCOCOg | |
|---|---|---|---|---|---|---|---|---|---|---|---|
| | | | | val | testA | testB | val | testA | testB | val | test |
| *Trained from scratch:* | | | | | | | | | | | |
| MCN$_{(CVPR20)}$ [54] | Darknet53 | bi-GRU | ✗ | 62.44 | 64.20 | 59.71 | 50.62 | 54.99 | 44.69 | 49.22 | 49.40 |
| EFN$_{(CVPR21)}$ [12] | ResNet101 | bi-GRU | ✗ | 62.76 | 65.69 | 59.67 | 51.50 | 55.24 | 43.01 | - | - |
| BUSNet$_{(CVPR21)}$ [82] | DeepLab-R101 | Self-Att | ✗ | 63.27 | 66.41 | 61.39 | 51.76 | 56.87 | 44.13 | - | - |
| CGAN$_{(MM20)}$ [53] | DeepLab-R101 | bi-GRU | ✗ | 64.86 | 68.04 | 62.07 | 51.03 | 55.51 | 44.06 | 51.01 | 51.69 |
| ISFP$_{(MM22)}$ [41] | Darknet53 | bi-GRU | ✗ | 65.19 | 68.49 | 62.73 | 52.70 | 56.77 | 46.39 | 52.67 | 53.00 |
| LST$_{(CVPR21)}$ [22] | Darknet53 | bi-GRU | ✗ | 65.43 | 67.76 | 63.08 | 54.21 | 58.32 | 48.02 | 54.40 | 54.25 |
| ReSTR$_{(CVPR22)}$ [24] | ViT-B | Transformer | ✗ | 67.22 | 69.30 | 64.45 | 55.78 | 60.44 | 48.27 | - | - |
| LAVT$_{(CVPR22)}$ [83] | Swin-B | BERT | ✗ | 72.73 | 75.82 | 68.79 | 62.14 | 68.38 | 55.10 | 61.24 | 62.09 |
| VLT$_{(TPAMI22)}$ [10] | Swin-B | BERT | ✗ | 72.96 | 75.96 | 69.60 | 63.53 | 68.43 | 56.92 | 63.49 | 66.22 |
| VDP$_{(ICCV23)}$ [89] | VQGAN | CLIP | ✗ | 73.25 | - | - | 62.69 | - | - | 61.96 | - |
| ReLA$_{(CVPR23)}$ [40] | Swin-B | BERT | ✗ | 73.82 | 76.48 | 70.18 | 66.04 | 71.02 | 57.65 | 65.00 | 65.97 |
| SADLR$_{(AAAI23)}$ [84] | Swin-B | BERT | ✗ | 74.24 | 76.25 | 70.06 | 64.28 | 69.09 | 55.19 | 63.60 | 63.56 |
| CARIS$_{(MM23)}$ [47] | Swin-B | BERT | ✗ | 75.14 | 78.10 | 71.75 | 66.63 | 72.04 | 58.51 | 64.31 | 65.82 |
| *Pre-trained:* | | | | | | | | | | | |
| MaLT$_{(ARXIV21)}$ [37] | ViLT | BERT | 3.88M | 70.13 | 71.71 | 66.92 | 62.23 | 65.92 | 56.06 | 62.45 | 62.87 |
| CRIS$_{(CVPR22)}$ [69] | CLIP-R101 | CLIP | 3.88M | 70.47 | 73.18 | 66.10 | 62.27 | 68.08 | 53.68 | 59.87 | 60.36 |
| ETRIS$_{(ICCV23)}$ [80] | CLIP-R101 | CLIP | 3.88M | 71.06 | 74.11 | 66.66 | 62.23 | 68.51 | 52.79 | 60.28 | 60.42 |
| SeqTR$_{(ECCV22)}$ [92] | Darknet53 | bi-GRU | 6.10M | 71.70 | 73.31 | 69.82 | 63.04 | 66.73 | 58.97 | 64.69 | 65.74 |
| SEEM$_{(NeurIPS23)}$ [93] | DaViT-d5 | Florence | 5.00M | - | - | - | - | - | - | 65.70 | - |
| LISA-7B$_{(ARXIV23)}$ [30] | ViT-H | LLaVA-7B | 18.16M | 74.90 | **79.10** | 72.30 | 65.10 | 70.80 | 58.10 | 67.90 | **70.60** |
| PolyFormer$_{(CVPR23)}$ [44] | Swin-L | BERT | 5.70M | 75.96 | 78.29 | 73.25 | **69.33** | **74.56** | **61.87** | **69.20** | 70.19 |
| *Layer-wise DIT$_B$* (Ours) | ViT-B | BERT | ✗ | 71.98 | 74.51 | 68.77 | 59.97 | 65.52 | 51.72 | 60.18 | 61.15 |
| *Layer-wise DIT$_H$* (Ours) | ViT-H | BERT | ✗ | **76.18** | 78.13 | **73.27** | 68.00 | 71.77 | 60.04 | 67.37 | 67.92 |

not that helpful for layer-wise DIT, on the contrary, reducing performance greatly. This result may suggests that the text words are well projected onto each layer of SAM with independent projectors. The combination with parameterized tokens somewhat declines their semantics. These results also indicate the difference between DITs and prompt tuning, *i.e.*, one is for semantic projection while the other is for task adaption.

*4.3.3 Different text word injections.* In our layer-wise DIT, we project the text prompts directly onto the visual space of each SAM's layer. Here, we also test DIT with alternative manners in Tab. 3, such as *cross-attention* and *cross-attention+FFN*. Interestingly, these complex methods do not further enhance SAM's text-aware capability, but decline performance to some extent. This case suggests that with the strong dependency modeling of SAM's ViT encoder, a direct semantic projection can already facilitate cross-modal alignment, well confirming the motivation of our DIT.

*4.3.4 The impact of tuning on SAM.* In Tab. 4, we further examine the default capability of SAM with point and box guidances on the RefCOCO benchmark. Here, we use the ground-truth boxes of RefCOCO for box-guided SAM, and randomly sample 5 points

on the ground-truth mask for point-guided segmentation. From Tab. 4, we can first see that on RefCOCO, the default SAM is still likely to segment the incorrect regions of the referent with ground-truth boxes. To explain, RefCOCO is a challenging benchmark, of which images often involve dense visual semantics, and a large box will make SAM easy to segment irrelevant region. In contrast, the point sampled from the object offers a better guidance, but still lags behind SOTA performance of RIS [44, 47]. Meanwhile, the joint tuning with all types of prompts can further improve the performance of SAM on following boxes and points. We also notice that our DIT method can help SAM achieve strong text-guided performance that is already close to the box-guided capability, well validating the effectiveness of our method. In terms of the inference speed, since the number of text tokens is often about 15-20, much fewer than that of image patches, the overall impact of DITs on inference time is limited.

*4.3.5 Comparison with the State-of-the-art.* In Tab. 5, we compare our L-DIT against the *state-of-the-art* (SOTA) methods of referring image segmentation on RefCOCO [86], RefCOCO+ [86], and RefCOCOg [56] datasets using the *oIoU* metric. Specially, on RefCOCO dataset, we outperforms existing SOTA methods trained

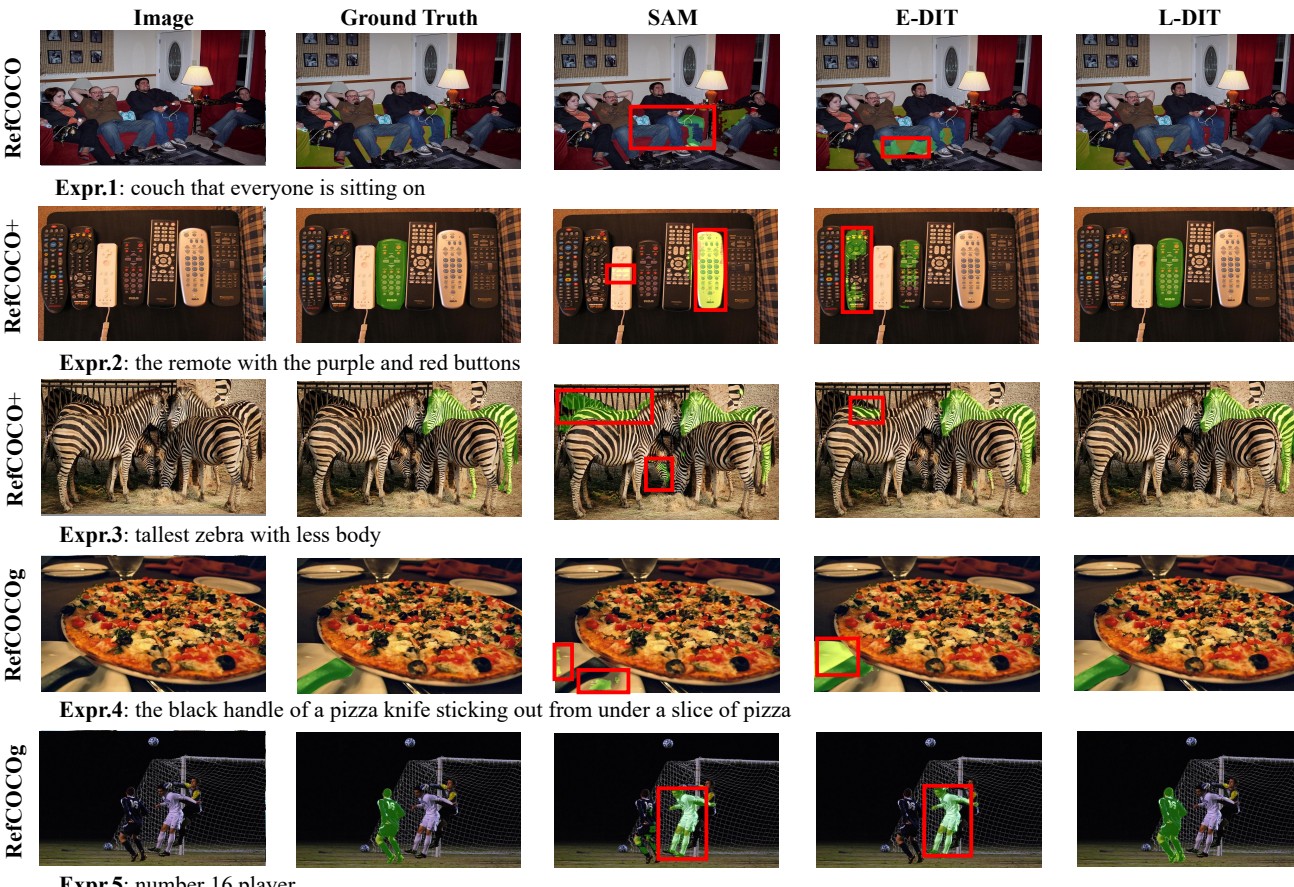

**Figure 4: Predictions of the default SAM and our end-to-end and layer-wise DITs, *i.e.*, E-DIT and L-DIT. Both methods can follow text instructions better than the default SAM, while L-DIT can accurately segment the referent even with complex text content or image background. For better comparison, we use red boxes to ground the incorrect segmentations.**

from scratch with absolute improvements of 1.94%, 1.56%, 2.91% in all three splits and performs on par with the pretrained Poly-Former [44]. Furthermore, on RefCOCO+ and RefCOCOg datasets, the L-DIT$_H$ also exhibits exceptional performance, demonstrating comparable capabilities to SOTAs [44, 84]. In addition, compared with SOTA RIS methods using a large number of visual grounding data for pre-training, Our L-DIT$_H$ is very competitive and achieves new SOTA performance on RefCOCO *val.*

### 4.4 Qualitative Analysis

To gain deep insights into DIT, we visualize the predictions of our DIT tuning methods and the default SAM in Fig.4. In the first column, it becomes evident that the default SAM struggles with a nuanced understanding of text, often yielding predictions that inadequately encapsulate the referent or outright incorrect. In contrast, the simple E-DIT offers a notable improvement of SAM's text comprehension capabilities. For instance, in the third example, SAM's segmentation exhibit a pronounced bias towards the distracting elements on the left, while E-DIT rectifies this issue, resulting in more accurate prediction. Notably, through E-DIT enhances the performance in some cases, it can lead to the loss of certain semantic information embedded within the textual features. As a

consequence, E-DIT occasionally misinterpret the segmentation target, as illustrated in the last example. The last column reveals that L-DIT effectively mitigates these issues without necessitating specific designs for cross-modal features. Remarkably, even without providing explicit image positions, akin to the original SAM, L-DIT consistently delivers superior segmentation results. This demonstrates that L-DIT processing can effectively harness positional information from both modalities and maintain a strong degree of context-awareness, improving accuracy and robustness.

### 5 Conclusion

In this paper, we propose two *deep instruction tuning* (DIT) methods to address the weakness of SAM on following text instructions, one is end-to-end and the other is layer-wise. DITs regard SAM's default visual encoder as a stand-alone multi-modal learner instead of adding additional fusion branches. Both approaches contribute to enhancing SAM's ability of segmenting target objects based on textual guidance. To validate DITs, we conduct extensive experiments on three RIS benchmark datasets. Experimental results show that our simple DIT methods can greatly improve the capability of SAM. Meanwhile, the proposed layer-wise DIT can even help SAM compete with a set of SOTA methods of RIS on all three datasets.

## Acknowledgments

This work was supported by National Science and Technology Major Project (No. 2022ZD0118201), the National Science Fund for Distinguished Young Scholars (No.62025603), the National Natural Science Foundation of China (No. U21B2037, No. U22B2051, No. 623B2088, No. 62176222, No. 62176223, No. 62176226, No. 62072386, No. 62072387, No. 62072389, No. 62002305 and No. 62272401), and the Natural Science Foundation of Fujian Province of China (No.2021J01002, No.2022J06001).

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
