# OpenReview forum: "Deep Instruction Tuning for Segment Anything Model"
_acmmm.org/ACMMM/2024/Conference — MM2024 Poster_

### Official Review · Reviewer_48Wd · 2024-05-23

**Rating:** 3
**Confidence:** 3

**Summary:**

The paper proposes the Deep Instruction Tuning (DIT) for SAM, to improve the performance of SAM on text instruction segmentation tasks. In order to give SAM the ability to understand text prompts, this article enhances SAM into a visual-language model.

**Strengths:**

The method is simple and intuitive, and the paper is clearly stated.

The ablation study is comprehensive.

**Limitations:**

1. The novelty is a bit low. The proposed method appears just to combine $F_t$ and $F_v$ for attention.

2. The proposed method does not exhibit sufficient generalization, as it is trained and tested only on the closed set. Additionally, it is necessary to demonstrate its performance in complex multi-target scenarios.

3. The proposed method does not demonstrate a significant advantage over SOTA methods, and the SOTA methods' results are also not shown in the visualizations.

**Suitability:**

3

---

### Official Review · Reviewer_qobo · 2024-05-24

**Rating:** 3
**Confidence:** 4

**Summary:**

This paper proposes to do deep text instruction tuning for SAM. They concatenate text tokens and image tokens together to get a more accurate text-prompt segmentation result.

**Strengths:**

1. Extensive experiments were conducted to fully evaluate the results of the method, including a large number of ablation studies.
2. Very well written. The article is easy to read.

**Limitations:**

1. From Table 7, we can find that L-DIT_B cannot surpass other training from scratch methods with the same level vision backbone Swin-B. The authors need to explain why they are lagging behind other SOTA work. Or provide speed test results to prove they have an advantage at the same speed.
2. The methods are too simple and natural, and the innovation cannot meet the requirements of a top-level conference.

**Suitability:**

2

---

### Official Review · Reviewer_vWZU · 2024-05-25

**Rating:** 5
**Confidence:** 4

**Summary:**

To enhance the text-prompt segmentation of SAM, this paper proposed two deep instruction tuning (DIT) methods. The author argued that the shallow fusion scheme in original SAM can not handle the multi-modal ability well, and it only interact to a limited extent with visual tokens via two cross-attention layers in its mask decoder. Inspired by the previous prompt tuning methods, this work introduced End-to-end DIT(E-DIT) and Layer-wise DIT(L-DIT) approaches. Extensive experiments demonstrate the effectiveness of the method and show the advantages of DIT-SAMs over SAM.

**Strengths:**

1. The proposed DIT keeps the SAM architecture unchanged and directly uses the image encoder as a multi-modal fusion network.

2. DIT is simple yet effective and the experimental results show the performance advantages.

3. Providing box- and point-prompted segmentation and  the inference time comparsions with original SAM.

**Limitations:**

1. Although it is effective compared to SAM, while the performance is not strong enough, compared to PolyFormer and LISA-7B.

2.Can the method segment multiple objects? Additional visual results would be valuable for showcasing the model's capabilities.

**Suitability:**

3

---

### Official Review · Reviewer_yW12 · 2024-05-26

**Rating:** 3
**Confidence:** 3

**Summary:**

The paper targets at improving the text to mask (more formally, referring segmentation) capability of Segment Anything Model (SAM). Similar to prompt tuning, two straightforward instruction tuning methods, either end-to-end and layer wise is proposed for SAM. The novelty in design is mainly aligning the text tokens with SAM image space before conditioning the mask decoder. The experiments show effectiveness of the instruction tuning on SAM and performance improvement on major referring segmentation datasets.

**Strengths:**

\textbf{Simple and Effective}: Using deep instruction tuning to align text space and SAM space is a good and (somewhat incremental) novel idea to enhance SAM’s performance in referring segmentation tasks. The intuitive configurations of layer-wise and end-to-end tuning are straightforward and effective.

\textbf{Detail Experiment and Ablation} The experiments are conducted extensively with two vision backbones and evaluated on various accuracy metrics. The results are strong and comparable to previous baselines. Comprehensive ablation studies validate the contribution of each component.

**Limitations:**

The authors claim that deep text instruction tuning is essential for SAM and reports less than 60 oIOU of Default SAM on RefCOCO as a primary motivation. However, there is no official implementation of SAM with text encoding capabilities and no extra explanation in this paper. Could you elaborate more on the implementation of SAM you use?

Using a pretrained CLIP feature without alignment could intuitively result in not good performance. Refer to some insights in
https://github.com/facebookresearch/segment-anything/issues/4

Therefore, comparing an instruction-tuned implementation with a SAM implementation that uses a pretrained CLIP encoder without additional alignment or training seems unfair. The statement that "Deep text instruction tuning is essential for SAM" is unconvincing without further evidence. In this way, these findings might be more suitable for a technical report rather than a conference paper.

I could be convinced in the rebuttal and increase the ratings if the authors address my concern towards motivation and defend research novelty well.

**Suitability:**

2

---

### Meta-Review · Area_Chair_JPkZ · 2024-07-02

**Recommendation:** Accept (Poster)
**Confidence:** 4

**Metareview:**

This paper proposes Deep Instruction Tuning (DIT) methods to enhance the Segment Anything Model's (SAM) capabilities in text-instructed image segmentation tasks.

Initial concerns mainly lie on the novelty of the proposed method, and the fairness of comparisons.
The authors provided a thorough response addressing the concerns raised by the reviewers, leading to a final scores of two Borderline Reject, one Borderline Accept and one Weak Accept, with an average score of 3.75. As two reviewers still had doubts about the novelty of the paper, the AC carefully reviewed the paper and all the reviewers' comments.

Considering the good performance and potential impact for the SAM community,the AC agrees to accept the paper and strongly recommends incorporating the content from the rebuttal into the final version.